ecology

organic agriculture, adaptation, *Daphnia magna*, deltamethrin, resistance, evolution

**Author for correspondence:**
Rafaela A. Almeida
e-mail: rafaela.almeida@kuleuven.be

# Differential local genetic adaptation to pesticide use in organic and conventional agriculture in an aquatic non-target species

Rafaela A. Almeida[1], Pieter Lemmens[1,2], Luc De Meester[1,2,3,4] and Kristien I. Brans[1]

[1]Laboratory of Aquatic Ecology, Evolution and Conservation, KU Leuven, Ch. Deberiotstraat 32, B-3000 Leuven, Belgium
[2]Leibniz Institute für Gewasserökologie und Binnenfischerei (IGB), Müggelseedamm 310, 12587 Berlin, Germany
[3]Institute of Biology, Freie Universität Berlin, Königin-Luise-Strasse 1-3, 14195 Berlin, Germany
[4]Berlin-Brandenburg Institute of Advanced Biodiversity Research (BBIB), Königin-Luise-Str. 2-4, 14195 Berlin, Germany

RAA, 0000-0002-5228-9091; PL, 0000-0002-3135-9724; LDM, 0000-0001-5433-6843; KIB, 0000-0002-0464-7720

Pesticide application is an important stressor to non-target species and can profoundly affect ecosystem functioning. Debates continue on the choice of agricultural practices regarding their environmental impact, and organic farming is considered less detrimental compared to conventional practices. Nevertheless, comparative studies on the impacts of both agricultural approaches on the genetic adaptation of non-target species are lacking. We assessed to what extent organic and conventional agriculture elicit local genetic adaptation of populations of a non-target aquatic species, *Daphnia magna*. We tested for genetic differences in sensitivity of different *D. magna* populations ($n = 7$), originating from ponds surrounded by conventional and organic agriculture as well as nature reserves, to pesticides used either in conventional (chlorpyrifos) or organic agriculture (deltamethrin and copper sulfate). The results indicate that *D. magna* populations differentially adapt to local pesticide use. Populations show increased resistance to chlorpyrifos as the percentage of conventional agriculture in the surrounding landscape increases, whereas populations from organic agriculture sites are more resistant to deltamethrin. While organic agriculture is considered less harmful for non-target species than conventional, both types of agriculture shape the evolution of pesticide resistance in non-target species in a specific manner, reflecting the differences in selection pressure.

## 1. Introduction

Current agriculture strongly relies on the use of agrochemicals, including pesticides [1,2]. However, such practice represents a particularly concerning threat to natural communities. Pesticides can move from the application site and enter the surrounding terrestrial and aquatic ecosystems via runoff and spray drift [3–5], and thereby affect non-target species and impact ecosystem structure and functioning [6–9]. For example, pesticide exposure was shown to affect macroinvertebrate community composition and decrease leaf-litter decompositions in streams in Europe [9].

Pesticide exposure can have lethal [10] as well as sub-lethal effects, such as phenotypic changes in life-history traits [11], behaviour [12] and physiology [13]. For instance, exposure to chlorpyrifos (CPF) decreased *Daphnia carinata* survival and, at lower concentrations, negatively affected reproductive parameters, like timing until first brood production and number of offspring [11]. However, populations can show increased tolerance to pesticides, achieved

either via phenotypic plasticity or via genetic adaptations. In the presence of substantial genetic variation, populations exposed to pesticides can genetically adapt to the toxicant via selection for the most tolerant individuals and an increase in resistant allele frequencies [14,15]. For instance, *Gammarus pulex* amphipods from agricultural streams evolved a considerably higher resistance to clothianidin (neonicotinoid) compared to amphipods from pristine streams [16]. However, several studies have also reported that increased resistance to pesticide can entail fitness costs to individuals [17,18], which may compromise the stability of ecosystems and ecosystem services provisioning.

In an effort to address the environmental concerns associated with pesticide applications in conventional agriculture, organic farming has been increasingly presented as a more benign alternative. Organic agriculture has been highly promoted under the EU Common Agricultural Policy (2014–2020), as it relies on the use of organic fertilizers and a restricted set of pesticides [19]. Most commonly applied pesticides in organic agriculture are compounds extracted from natural sources, such as pyrethrins, with the exception of some synthetic products such as copper sulfate (CS) and some pyrethroids. Nevertheless, the toxicity of pesticides used in organic agriculture can be comparable to that of conventional pesticides [20,21]. So far, no studies have tested whether organic and conventional pesticide use differentially drive genetic adaptation in life history, behaviour or physiology of non-target species.

Ponds are abundant ecosystems in agricultural landscapes, and substantially contribute to regional biodiversity [22] and ecosystem services [23,24]. Their ecological quality is largely determined by local surrounding land use [25]. Large-bodied zooplankton, such as *Daphnia magna* and *Daphnia pulex*, are particularly valuable organisms in pond communities as they play a key role in pond food webs by exerting top-down control on primary producers, and in turn being an important food source for macro-invertebrates and vertebrates [26]. Their fast generation time and the possibility to work with clones (i.e. from cyclical parthenogenesis) make *Daphnia* a very suitable model organism in ecotoxicology [27,28], ecology and evolutionary biology [26,29,30]. *Daphnia* species are generally common in farmland ponds and are therefore frequently exposed to pesticides. Earlier studies reveal that this can lead to evolved pesticide resistance [31,32]. For example, intensive agriculture around ponds leads to evolved increased tolerance to carbaryl (carbamate) in *D. magna* populations [32]. While evolutionary responses to conventional pesticides are ubiquitously assessed across a large range of non-target animals [18,31,33,34], it remains unanswered to what extent these responses compare to those caused by pesticides allowed in organic agriculture. There is a need to assess whether and how organic agriculture drives adaptive pesticide resistance in non-target populations, in order to better evaluate the potential impact of organic agriculture on wildlife.

We tested for differential genetic adaption of *D. magna* populations to pesticides used in conventional and organic farming using a laboratory common garden toxicity assay. Five clonal lineages of seven *D. magna* populations from agricultural areas with either of the two pesticide application strategies, or from non-agricultural natural sites, were exposed to a range of concentrations of both conventional (CPF) and organic pesticides (deltamethrin, hereafter DTM; CS). We assessed median effective concentrations for each pesticide across all clones and populations. We hypothesize that *Daphnia* populations differ in their tolerance to pesticides

to match the pesticide strategy applied in the surroundings of the ponds from where they originate. More specifically, we predict populations from ponds in agricultural sites under conventional pesticide management to be more resistant to CPF compared to populations from organic farmlands or nature reserves. Conversely, we expect that populations in organic farmland have evolved a higher resistance to CS and DTM compared to populations from areas of conventional agriculture and nature reserves. We predict populations from ponds in nature reserves to have the lowest overall pesticide tolerance.

## 2. Material and methods

### (a) *Daphnia magna* study populations and pre-experimental rearing conditions

We sampled seven *D. magna* populations from ponds located in Western Flanders (Belgium). Five ponds were located in agricultural farmland either under conventional (three ponds: 'C1', 'C2' and 'C3') or organic farming management (two ponds: 'O1' and 'O2'); two populations originated from ponds in nature reserves ('N1', 'N2') (electronic supplementary material, table S1). The percentage of area under organic and conventional agriculture in a perimeter of 200 m around each pond was determined by combining GIS data ('Watervlakken' blue layer; available at Geopunt Vlaanderen, https://www.geopunt.be/; [35]) with the annual agricultural parcel registration data of the Department of Agriculture and Fisheries (see electronic supplementary material, table S1). Land use in a perimeter of 200 m has been shown to be highly relevant with respect to the physico-chemical and biotic characteristics of small ponds [25]. We reared and identified five clonal lineages per population ($n = 7 \times 5 = 35$ clonal lineages) (see detailed information in electronic supplementary material, material and methods section).

Clonal lineages were maintained for at least six generations under standardized pre-experimental laboratory conditions (16 : 8 light : dark photoperiod, reared in dechlorinated tap water at $20 \pm 1$°C, and fed with the green algae *Acutodesmus obliquus* at a concentration of $1 \times 10^5$ cells ml$^{-1}$ twice per week). Subsequently, clones used in this experiment were grown under experimental standardized conditions for two generations to minimize the interference from (grand)maternal effects on experimental responses. Four replicated cultures of each clonal lineage were established using parthenogenetically produced offspring, with five individuals raised in 500 ml glass jars filled with 24 h aged dechlorinated tap water, and kept at $20 \pm 1$°C under a 16 : 8 light : dark photoperiod. Seventy per cent of the medium was refreshed every second day, after which food concentrations were restored to $1 \times 10^5$ cells ml$^{-1}$.

### (b) Pesticide acute toxicity experiments: pesticides and experimental design

#### (i) Pesticides

We established dose–response curves for all 35 clonal lineages for three different pesticides: DTM, CS and CPF. DTM (CAS 52918-63-5, purity greater than 98%, Sigma-Aldrich) is a synthetic pyrethroid insecticide with a similar chemical structure as that of pyrethrins [36]. DTM is used worldwide and together with lamba-cyhalothrin are the only pyrethroid compounds (biosynthesis via the plant *Tanacetum cinerariaefolium* [37]), that are allowed in organic agriculture under the EU Commission Regulation (EC) No 889/2008 [38]. Pyrethroids have the same neurotoxic properties as pyrethrins [39]; they disrupt the

function of voltage-gated sodium channels, preventing the transition from an opened state (active) to a closed state (inactive) after the action potential has been initiated [36,40,41]. CS (pentahydrated) (CAS 7758-99-8, VWR international) is used in organic agriculture as a herbicide, fungicide, root killer, algaecide and in some cases a molluscicide and bactericide [42], and is known to accumulate in the environment [43]. Although copper is an essential element for most organisms, it can lead to oxidative stress, DNA damage, denaturation of proteins and deactivation of enzymes when present at high concentrations [44–46]. CPF (CAS 2921-88-2, purity greater than 99%, Sigma-Aldrich) is a broad-spectrum organophosphorus insecticide commonly used in conventional agriculture [47]. Organophosphorus insecticides act as acetylcholinesterase inhibitors, which evokes neurotoxic responses [27].

### (ii) Acute toxicity assays and median effective concentration (EC$_{50}$)

We followed the OECD 202 guideline for acute immobilization tests in *Daphnia sp.* [48]. Five juveniles (less than 24 h old) were inoculated in 100 ml vials filled with 50 ml of test solution. The individuals were not fed during the exposure and were kept at $20 \pm 1°C$ and 16 : 8 light : dark photoperiod. We quantified the number of immobilized individuals after 48 h by gently agitating the jar and checking for movement for 15 s. We scored immobilization for all 35 clones when exposed to seven concentrations of each pesticide (DTM: 0.04–2.56 µg l$^{-1}$, CS: 17.675–1131.2 µg l$^{-1}$ and CPF: 0.094–6 µg l$^{-1}$, electronic supplementary material, table S2) plus a solvent control. Each clone × pesticide × concentration combination was replicated three times, with replicates originating from three out of the four original replicate jars (individually cultured lineages) of each clonal lineage available, and all replicate × pesticide concentration combinations were randomized through time (electronic supplementary material, figure S1).

The chosen ranges for each pesticide were based on the Pesticide Action Network database (see http://www.pesticideinfo.org/ and https://cfpub.epa.gov/ecotox/index.cfm). We selected an initial environmentally relevant concentration, which was then compared with EC$_{50}$ values from the literature. For CPF and DTM, we ultimately selected a published EC$_{50}$ value (48 h exposure) that was derived from a methodologically strong study. More specifically, for CPF, Palma *et al.* [49] reported an EC$_{50}$ of 0.74 µg l$^{-1}$ for *D. magna*. For DTM, the concentrations were chosen based Toumi *et al.* [50], reporting EC$_{50}$ values of 0.32 µg l$^{-1}$ and 0.63 µg l$^{-1}$ for two different *D. magna* strains. In the case of CS, the concentrations (EC$_{50}$ 141.4 µg l$^{-1}$) were selected after a short pilot experiment that allowed us to correct our previously selected concentrations based on literature (EC$_{50}$ 10.5–70.7 µg l$^{-1}$ [51]) because these had proven too low to detect a response in our clones. The selected EC$_{50}$ values for each pesticide was accompanied by three higher and three lower concentrations, each time obtained by either multiplying or dividing the flanking concentrations by two (electronic supplementary material, table S2).

### (c) Statistical analyses

All analyses were performed in R v. 4.0.2 [52].

Median effective concentrations (EC$_{50}$) for CPF, DTM and CS were estimated by a four-parameter log-logistic model using the 'drc' package [53], resulting in four response parameters: the slope, lower limit, upper limit and EC$_{50}$ of the dose–response curve. EC$_{50}$ values were used in the statistical analyses.

Considering pond selection, the amount of organic agriculture in the immediate surroundings of the ponds was either very low (0%, for conventional agriculture and natural ponds) or high (greater than 80%, for organic agricultural ponds). By contrast, the percentage of conventional agriculture varied

more gradually, from 0% (natural ponds) to greater than 75% (conventional agriculture), with organic farmland ponds showing intermediate (49% and 50%) amounts of conventional agriculture. Taking this small-scale heterogeneity into account in the analysis is important, as it is possible that the populations obtained from organic agriculture had a certain degree of exposure to the pesticide regime of conventional agriculture. In reverse, our populations from conventional agriculture have never been surrounded by organic farming and were thus likely not exposed to organic pesticides. Therefore, a categorical approach, in which populations are categorized as organic, conventional or natural, according to the surrounding local land use type is better suited to analyse the response to pesticides in organic farming, such as DTM and CS. To test whether populations exposed to organic farming showed different EC$_{50}$ values for certain pesticides than populations from conventional agriculture or nature reserves, we carried out a linear mixed-effect model with land use type as a fixed effect, and population (nested in land use type) and clone (nested in population) as random effects. For CPF, however, it is important to consider the land use around each pond as a gradient of increased intensity of conventional agriculture. To test for the effect of conventional agriculture on EC$_{50}$ values, we constructed a linear mixed-effect model including the percentage of conventional agriculture at 200 m radius around each pond as a continuous explanatory variable, with clone nested in population as a random effect. For transparency, we report on patterns observed in both analyses for all pesticides.

Model assumptions were checked visually by plotting residuals versus fitted values and normal Q–Q plots, and deviations from normality were formally tested with the Shapiro–Wilk test, both in the analyses using land use type (fixed effect) and percentage of land use types around the ponds (continuous explanatory variable). EC$_{50}$ of the three pesticides were log-transformed to better meet assumptions of normality. All linear mixed-effect models were fitted using the 'lme4' [54] package, and *p*-values and approximate F-test statistics were computed using the 'car' package [55]. We used restricted maximum-likelihood estimation method and corrected the degrees of freedom for fixed effects using the Kenward–Roger approximation. Pairwise comparisons were analysed using Tukey HS post hoc testing with the package 'multcomp'[56]. The significance of random effects was tested using the package 'lmerTest' [57].

## 3. Results

Specific model parameter estimates (slope, lower limit, upper limit, EC$_{50}$) after EC$_{50}$ estimation for each population group (nature, conventional, organic) and for each pesticide (CPF, DTM and CS) are given in electronic supplementary material, table S3, and visually presented in figure 1$a$,$c$,$e$.

### (a) Resistance to pesticides used in organic agriculture

In the categorical analyses, EC$_{50}$ for DTM (figure 1$a$,$b$) differed significantly between the land use types ($F_{2,4.01} = 9.971$, $p = 0.028$; electronic supplementary material, table S4). Populations located in agricultural landscapes with organic pesticide management showed a higher average EC$_{50}$ for DTM compared to those from populations inhabiting ponds in conventional agricultural sites or nature reserves, indicating a genetic increase in DTM resistance in these populations. Post hoc comparison revealed significant differences in response to DTM between organic and conventional agriculture (0.877 estimate difference, $p < 0.001$), and between organic agriculture and nature reserve (0.592 estimate difference, $p = 0.015$). The random effect of population did not significantly contribute to

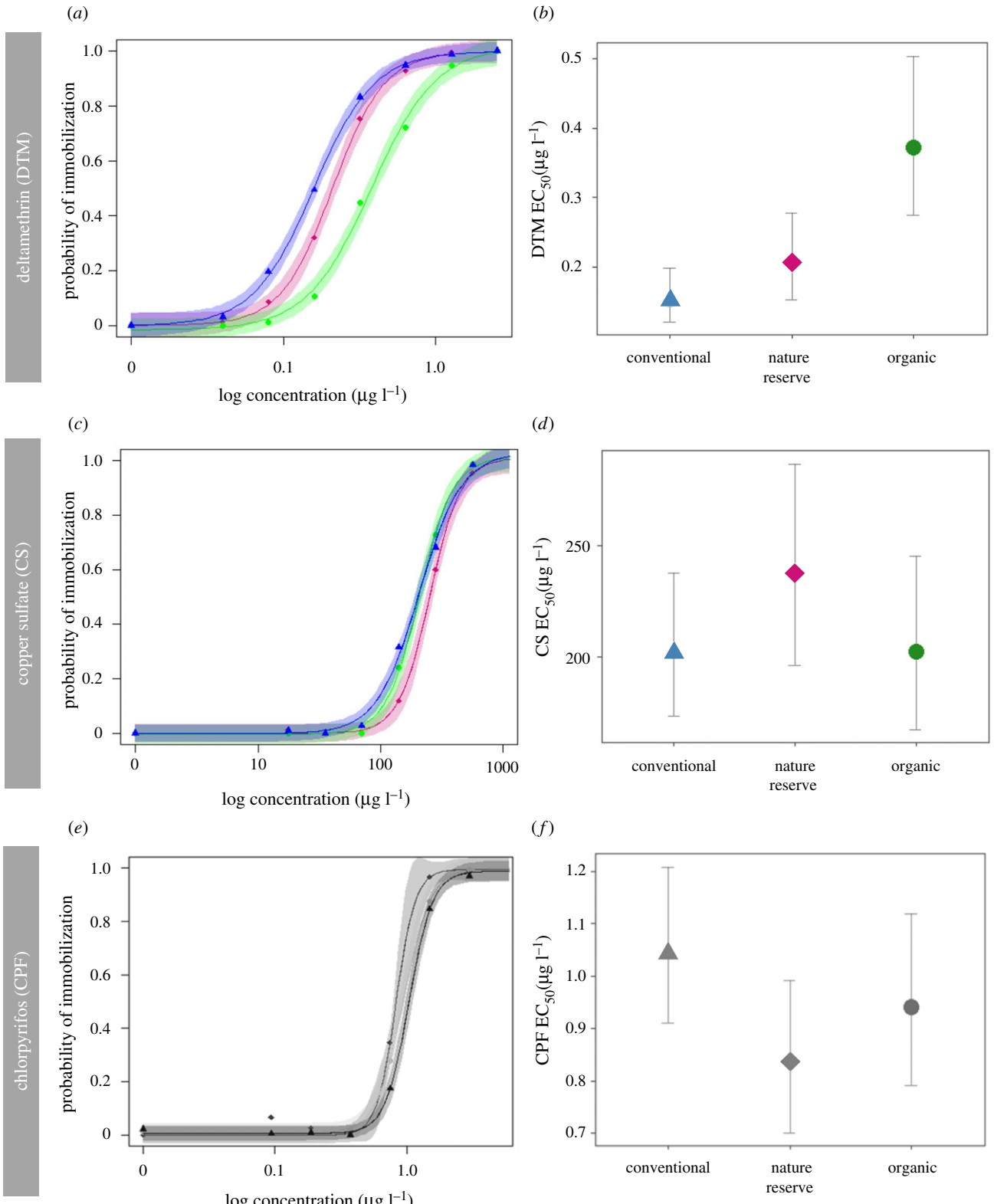

**Figure 1.** Logarithmic concentration (µg l$^{-1}$) – response (immobilization at 48 h) curves (*a,c,e*) and average EC$_{50}$ values (backtransformed) ± CI. (*b,d,f*), as assessed in a common garden rearing experiment using *D. magna*, for DTM (*a,b*), CS (*c,d*) and CPF (*e,f*) for each land use type (nature reserve: diamonds, conventional agriculture: triangles, organic agriculture: circles) in which the ponds, from which the *D. magna* study populations were isolated, were located. Shadowed areas in the left panels corresponds to 95% CI. (*a–d*) Panels represented in colour correspond to the pesticides allowed in organic agriculture (DTM and CS), and (*e,f*) panels in grey correspond to pesticide allowed in conventional agriculture (CPF). (Online version in colour.)

observed variation in resistance to DTM (d.f. = 1, *p* = 0.293; electronic supplementary material, table S4), nor did clone (d.f. = 1, *p* = 0.179).

The EC$_{50}$ values for CS did not differ between the land use types (main effect land use type, $F_{2,4.147}$ = 0.9231, *p* = 0.466; electronic supplementary material, table S4).

Population as random effect did not significantly contribute to observed variation in CS resistance (d.f. = 1, *p* = 0.922); neither did clone (d.f. = 1, *p* = 0.14).

Despite the fact that the gradual analysis is less ideal for testing the effect of organic agriculture in our study (see Material and methods), we still observed a significant effect

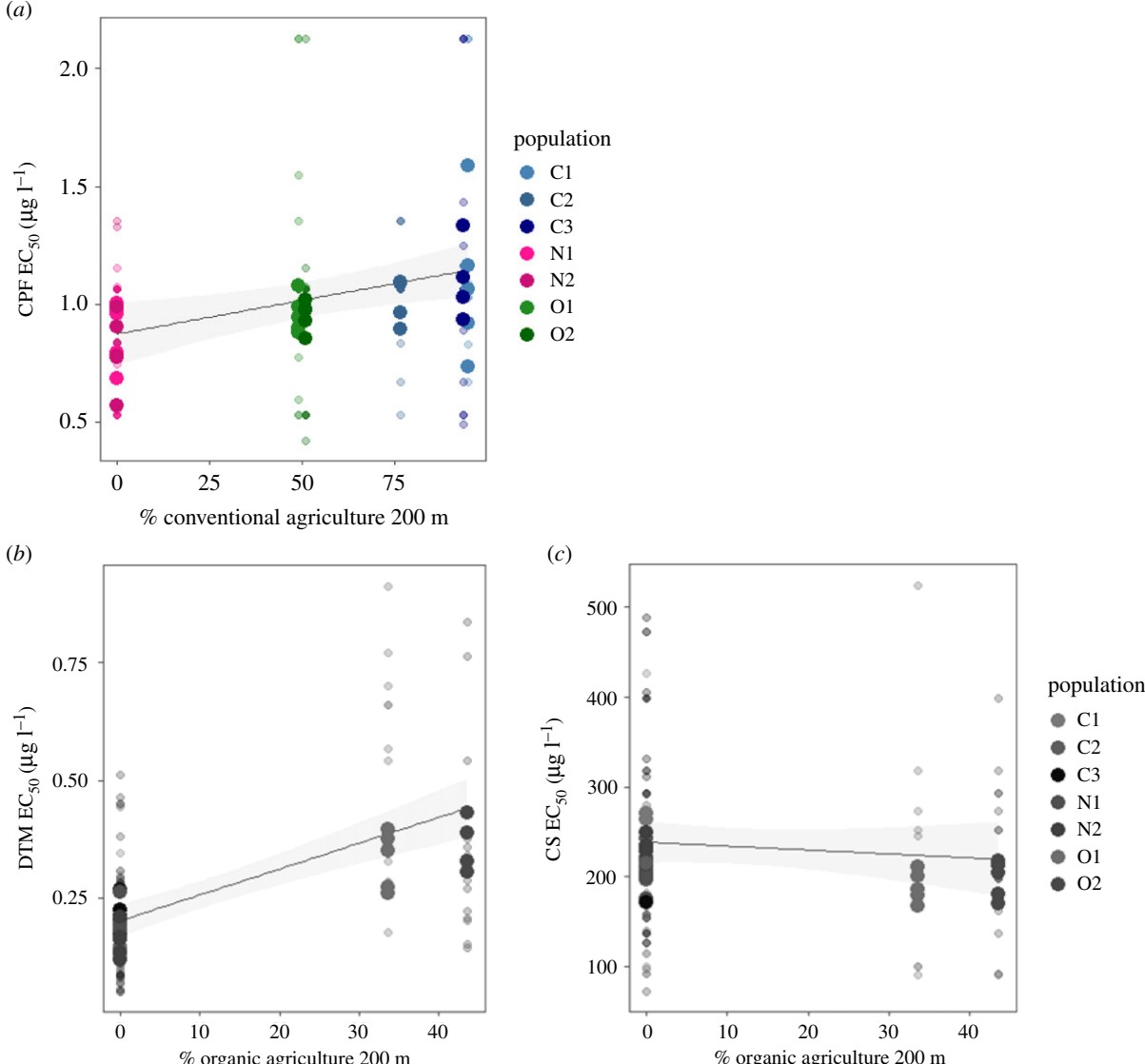

**Figure 2.** Pesticide resistance expressed as EC$_{50}$ (median effective concentration) in response to the percentage of agricultural land use type (conventional: (*a*), organic: (*b,c*)) within a 200 m radius around the ponds from which study populations of *D. magna* were collected. Categorical classifications of populations based on land use type are indicated in colour; nature reserves: pink, conventional agriculture: blue and organic agriculture: green. Large circles represent the mean EC$_{50}$ per clone in each population and small circles represent the EC$_{50}$ values of each replicate. (*a*) Panel represented in colour corresponds to the pesticides used in conventional agriculture (CPF), and (*b,c*) panels in grey correspond to pesticides used in organic agriculture (DTM and CS). (Online version in colour.)

of organic agriculture on the EC$_{50}$ values for DTM, with an increase in tolerance with increasing amounts of organic agriculture in a perimeter of 200 m around the ponds ($F_{1,33.737} = 24.465$, $p < 0.001$; electronic supplementary material, table S5; figure 2*b*). The random effect of clone (nested in population) was significant (d.f. = 1, $p = 0.015$; electronic supplementary material, table S5). By contrast, resistance to CS did not correlate with the amount of organic agriculture in the surroundings of the pond ($F_{1,32.944} = 0.291$, $p = 0.593$) (figure 2*c*). The random effect of clone (nested in population) was not significant (d.f. = 1, $p = 0.11$).

## (b) Resistance to pesticides used in conventional agriculture

The results for the linear mixed model relating EC$_{50}$ of CPF to the amount of conventional agriculture revealed a significant increase of resistance to CPF along with a gradient of increasing amounts of conventional agriculture in the vicinity of the pond ($F_{1,33} = 4.38$, $p = 0.044$; electronic supplementary material, table S5; figure 2*a*). For this model, the random

effect clone (nested in population) was significant (d.f. = 1, $p = 0.003$; electronic supplementary material, table S5), indicating overall among-clonal variation in CPF resistance across populations.

For the linear mixed model using categorical land use type (conventional, organic, nature reserve), *D. magna* clones obtained from conventional agriculture farms did not show higher EC$_{50}$ for CPF compared to the individuals obtained from organic farms and natural reserves ($F_{2,32} = 2.097$, $p = 0.139$; electronic supplementary material, table S4; figure 1*e,f*). The population did not significantly contribute to the observed variation in CPF tolerance in this model and was omitted from further analysis for reasons of model simplification. There was, however, significant variation among clones within populations (d.f. = 1, $p = 0.02$).

## 4. Discussion

We investigated the genetic adaptation of *D. magna* to pesticide applications in organic and conventional agriculture.

Using acute toxicity assays, we assessed *D. magna* resistance to the pesticides DTM and CS, allowed in organic agriculture, and CPF, one of the pesticides most commonly associated with conventional agriculture worldwide [58,59]. We found differential genetic adaptation of *D. magna* populations to both agricultural practices. Specifically, across all studied populations, populations from ponds located in organic agriculture sites evolved the highest resistance to DTM. Similarly, we observed a higher resistance to CPF in populations with the increasing amount of conventional agriculture near the pond. No signal of genetic adaptation was found for CS. Our results reveal highly heterogeneous pesticide adaptation to the specific local agricultural practice type in a non-target freshwater species and highlight the capacity of non-target key interactors to genetically adapt to not only conventional, but also organic agriculture practices.

There is mounting evidence that *Daphnia* can adapt to global change drivers, including climate change [60,61], urbanization [62], metal pollution [63] and the application of conventional pesticides [32,64]. Pesticide use from conventional agriculture was shown to drive genetic adaptation in several other organism groups, including macro-invertebrates [16], mosquitos [65] and amphibians [33,34]. Our results on the adaptation of *D. magna* to CPF are in line with earlier studies on *D. magna* by Coors *et al.* [32] and Jansen *et al.* [31]. However, the increased conversion from conventional to organic agricultural practices under the EU Common Agricultural Policy raises the unanswered questions of whether and how the application of pesticides allowed in organic agriculture similarly shapes genetic adaptation in non-target organisms. The selected organic farms from which our clones were isolated have been under organic agricultural practices for 8 (population O1) and 9 years (population O2), meaning that, within this period, the populations residing in those ponds were able to evolve resistance to DTM.

Although CS is a pesticide commonly used in organic agriculture [21], no differences were found between the resistance of the populations from the different types of land use and thus no genetic adaptation in populations from organic farmlands. When compared to reported 48 h $EC_{50}$ values of CS for *D. magna* (9.4–140 µg l$^{-1}$) [66–71], the values obtained in our study are higher, for all three land use types (221–254 µg l$^{-1}$), implying a higher resistance to CS. One important aspect to consider is that copper shows speciation in freshwater environments [72–74]. This can affect the bioavailability of copper in the water column. Small variations in water chemistry between studies, such as pH, can affect this speciation process, and this can lead to differences in sensitivity [75]. Another possible explanation for this is the presence of copper in the soils across the larger region of our study area (see European Commission report EUR 27607 EN [76]). Although effective concentrations of copper can depend on the soil type [77,78], there is a high abundance of different sources of copper contamination in Flanders, including agricultural manuring [78], and also, particularly in Western Flanders, residues from First World War weaponry [79]. It is, therefore, plausible that the *Daphnia* populations included in this study have had high exposure to copper, resulting in evolved overall higher resistance to CS across all populations. Resurrection ecology [80–82], in which dormant *D. magna* resting stages from the past could be hatched and compared to current populations for their resistance to CS, could shed new light on this, but was beyond the scope of our study.

While organic agriculture relies on a restricted set of pesticides of a more natural origin [19], our results show that they can affect natural biota and non-target species equally strong as those of conventional farming. Some authors argue that, despite being derived from natural compounds, pesticides allowed in organic agriculture can be highly toxic [20,83,84]. Our results show that an increase of land dominated by conventional agriculture translates into an increased resistance to CPF in *D. magna* populations, whereas being surrounded by organic agriculture translates into an increased resistance to DTM. Our study is based on a limited number of populations, but clearly indicates that adaptive evolution in response to organic agriculture does occur, similarly to observations for conventional agriculture. In addition, it should be emphasized that one can only quantify genetic adaptation in populations that persist, whereas studies like ours do not show to what extent populations and species disappeared from the same habitats because they could not adapt, or not quickly enough. Organisms with longer generation times and lower evolutionary potential might fail to adapt to pesticides, or adaptation may take long time. Importantly, the adaptation to both pesticides seems to be independent. While cross-resistance to multiple pesticides has been observed, including in the flour beetle *Tribolium castaneum*, which showed resistance to organophosphorus as well as pyrethrins and pyrethroid insecticides [85], it was not observed in the current study (results section Cross-tolerance or trade-off in pesticide resistance, electronic supplementary material, figure S2). Briefly, the populations that displayed a higher CPF tolerance did not have a higher resistance to DTM, nor vice versa. This observation that there is no cross-tolerance for CPF and DTM and that DTM seems to be quite toxic indicates that, in terms of pesticides, the transition from conventional to organic agriculture might pose additional stress to non-target organisms.

While adaptive pesticide resistance can facilitate the short-term persistence of populations in the presence of a pesticide [32], it can entail a series of trade-offs that potentially impede long-term persistence in changing environments [17,18]. For instance, evolved carbaryl tolerance in *Daphnia* comes at a cost of a higher susceptibility to pathogens and parasites [17]. Furthermore, energy investment in detoxification mechanisms can imply a reduction of energy fluxes to important life-history traits, such as reproduction [86,87] and development time [88]. Moreover, clonal sorting and population size reductions during strong selection may erode the genetic diversity of the population and reduce effective population sizes, causing a risk for inbreeding [89], which can hinder the population's capability to adapt to other stressors. Cambronero *et al.* [90] showed that negative impacts of historical exposure to carbamate increased when combined with a rise of temperature. *D. magna* are key grazers in standing waters such as lakes and ponds [26]. Reduced population growth rates and densities of populations of large-bodied zooplankton grazers such as *Daphnia* might result in a decrease in the top-down control of phytoplankton growth [91,92] and thus increase the likelihood of formation of noxious blooms in eutrophying systems [93]. While rapid evolution in *D. magna* thus can enable persistence in landscapes with organic and conventional farming practices, the impact on ecosystem stability and functioning as a consequence of potential costs and trade-offs needs to be further assessed.

# 5. Conclusion

Our study demonstrates that *D. magna* populations isolated from ponds located in agricultural areas genetically adapt to the pesticides associated with the local agricultural management practices, more specifically conventional or organic agriculture. This local genetic adaptation translated into an increased resistance of *D. magna* from organic farms to DTM, and a positive correlation between tolerance to CPF and the amount of conventional agriculture near the ponds. This highlights the highly localized and heterogeneous impact of pesticide use from both conventional and organic agriculture on evolutionary processes in non-target species, which should be considered in our efforts to reduce the environmental footprint of agriculture.

Data accessibility. Data are available from the Dryad Digital Repository: https://doi.org/10.5061/dryad.pvmcvdnmh [94].

Authors' contributions. R.A.A.: conceptualization, data curation, formal analysis, methodology, visualization and writing-original draft; P.L.: formal analysis, funding acquisition, supervision, writing-review and editing; L.D.M.: conceptualization, formal analysis, funding acquisition, methodology, supervision, writing-review and editing; K.I.B.: formal analysis, methodology, supervision, writing-review and editing.

All authors gave final approval for publication and agreed to be held accountable for the work performed therein.

Competing interests. The authors declare no competing interests.

Funding. This work was supported by BELSPO (ORCA Project—BR/175/A1/ORCA). R.A.A. acknowledges a FWO PhD SB fellowship (application 1S04618 N). K.I.B. acknowledges a FWO Postdoctoral fellowship (application 1222120 N).

Acknowledgements. We thank the editor and two anonymous reviewers for their constructive comments and feedback on an earlier version of the manuscript. We thank Edwin van den Berg for culturing the algae, and Reindert Ekelson and Filipe Pinto for valuable help with practical work. We thank our partners in the Belspo project ORCA for logistical support.

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
