## [Peer Review File · Proceedings of the Royal Society B: Biological Sciences]

Review History

RSPB-2021-1903.R0 (Original submission)

Review form: Reviewer 1

Recommendation

Accept with minor revision (please list in comments)

Scientific importance: Is the manuscript an original and important contribution to its field?

Excellent

General interest: Is the paper of sufficient general interest?

Excellent

Quality of the paper: Is the overall quality of the paper suitable?

Excellent

Is the length of the paper justified?

Yes

Should the paper be seen by a specialist statistical reviewer?

No

Do you have any concerns about statistical analyses in this paper? If so, please specify them explicitly in your report.

No

It is a condition of publication that authors make their supporting data, code and materials available - either as supplementary material or hosted in an external repository. Please rate, if applicable, the supporting data on the following criteria.

Is it accessible?

Yes

Is it clear?

Yes

Is it adequate?

Yes

Do you have any ethical concerns with this paper?

No

Comments to the Author

Manuscript is well-written, the study is well-conceived, the data are thoroughly and appropriately analyzed and clearly presented, the results are interesting and the conclusions well-supported by the data. Importantly, the discussion is thoughtful, interesting, informative, and insightful. The findings convincingly demonstrate that, although "organic agriculture is considered less harmful for non-target species than conventional agriculture, both types of production" include the use of specific pesticides that select for resistance in non-target aquatic species. To my knowledge this is the first such report that directly and rigorously documents and compares such effects across conventional and organic production systems. The findings are novel and important to a complete understanding of the impacts of pesticide use in both conventional and organic systems and should be of broad interest, especially as they relate to ecological risk assessment, environmental toxicology, agroecology and crop protection.

I offer the following points for consideration:

Figure 1: I infer that the Y-axis values are predicted probabilities of immobilization. This should be specified.

Line 341: Not clear that mitigation is the appropriate term here.. Mitigating persistence of a population would seem to mean reducing the likelihood the population persists.

Review form: Reviewer 2

Recommendation

Accept with minor revision (please list in comments)

Scientific importance: Is the manuscript an original and important contribution to its field?

Excellent

General interest: Is the paper of sufficient general interest?

Excellent

Quality of the paper: Is the overall quality of the paper suitable?

Excellent

Is the length of the paper justified?

Yes

Should the paper be seen by a specialist statistical reviewer?

No

Do you have any concerns about statistical analyses in this paper? If so, please specify them explicitly in your report.

No

It is a condition of publication that authors make their supporting data, code and materials available - either as supplementary material or hosted in an external repository. Please rate, if applicable, the supporting data on the following criteria.

Is it accessible?

Yes

Is it clear?

Yes

Is it adequate?

Yes

Do you have any ethical concerns with this paper?

No

Comments to the Author

The study provides a nice example that pesticides used in organic and conventional agriculture affect genetic adaptation in terms of tolerance. Effects were more pronounced for deltamethrin though. The study is well performed. I found, however, some minor issues that can improve it. First I recommend to include in the SI a figure explaining the experimental set up. The typical ones showing populations, clones within populations, replicated within clones and how from each replicate test were performed. It is not clear to me the paragraph that describe that. If I am not wrong the authors mention that they perform acute test from each of the three replicates from each clone but in different times (1-3 days I guess, since it is impossible to keep synchronized 35 clonal lineages. I guess this is what authors mean to say. I diagram will improve that issue.

Other issue. When comparing in the discussion Cu EC50s across studies. Authors have to take into a count that Cu suffers from a complex speciation in fresh water, This means that usually only a very small fraction of the total Cu use in experiments (i.e. 3%) is bioavailable to animals. This means the very minor variation in the water used in different studies may affect dramatically EC50 for Cu. This is probably the reason justifying the apparently higher tolerance to Cu found in this study when compared with previous ones (Barata et al 1998).

It is also interesting to see that apparently deltamethrin induced greater changes in tolerance than chlorpyrifos. This may be related to the chemical properties and mode of action of both compounds and on the microevolutionary consequences. Pyrethroids have a KWO much greater than chlorpyrifos, which means that they can be adsorbed to particles and hence be more persistent in the environment (in sediments, edible food, etc). Chlorpyrifos just kill Daphnia, whereas pyrethroids at sublethal concentrations dramatically impair reproduction. This means that at low environmental concentrations pyrethroid insecticides are likely to affect to a greater extent Daphnia since they produce effects at lower concentrations than just chlorpyrifos and are more environmental persistent than chlorpyrifos. This is however, a conjecture, that need to be tested measuring pesticide levels on the suites and around then where Daphnia was collected

Decision letter (RSPB-2021-1903.R0)

05-Oct-2021

Dear Ms A. Almeida

I am pleased to inform you that your manuscript RSPB-2021-1903 entitled "Differential local genetic adaptation to pesticide use in organic and conventional agriculture in an aquatic non-target species" has been accepted for publication in Proceedings B.

The referee(s) have recommended publication, but also suggest some minor revisions to your manuscript. Therefore, I invite you to respond to the referee(s)' comments and revise your manuscript. Because the schedule for publication is very tight, it is a condition of publication that you submit the revised version of your manuscript within 7 days. If you do not think you will be able to meet this date please let us know.

It is a condition of publication that data supporting your paper are made available either in the electronic supplementary material or through an appropriate repository. Please see our Data Sharing Policies <https://royalsociety.org/journals/authors/author-guidelines/#data>.

Sincerely,

Dr Locke Rowe

Associate Editor

Board Member: 1

Comments to Author:

Congratulations -- two reviewers recommend your paper for publication and find your study well-conceived, well-written, and important. I invite you to make revisions according to the reviewers' (minor) suggestions.

Reviewer(s)' Comments to Author:

Referee: 1

Comments to the Author(s)

Manuscript is well-written, the study is well-conceived, the data are thoroughly and appropriately analyzed and clearly presented, the results are interesting and the conclusions well-supported by the data. Importantly, the discussion is thoughtful, interesting, informative, and insightful. The findings convincingly demonstrate that, although "organic agriculture is considered less harmful for non-target species than conventional agriculture, both types of production" include the use of specific pesticides that select for resistance in non-target aquatic species. To my knowledge this is the first such report that directly and rigorously documents and compares such effects across conventional and organic production systems. The findings are novel and important to a complete understanding of the impacts of pesticide use in both

conventional and organic systems and should be of broad interest, especially as they relate to ecological risk assessment, environmental toxicology, agroecology and crop protection.

I offer the following points for consideration:

Figure 1: I infer that the Y-axis values are predicted probabilities of immobilization. This should be specified.

Line 341: Not clear that mitigation is the appropriate term here.. Mitigating persistence of a population would seem to mean reducing the likelihood the population persists.

Referee: 2

Comments to the Author(s)

The study provides a nice example that pesticides used in organic and conventional agriculture affect genetic adaptation in terms of tolerance. Effects were more pronounced for deltamethrin though. The study is well performed. I found, however, some minor issues that can improve it. First I recommend to include in the SI a figure explaining the experimental set up. The typical ones showing populations, clones within populations, replicated within clones and how from each replicate test were performed. It is not clear to me the paragraph that describe that. If I am not wrong the authors mention that they perform acute test from each of the three replicates from each clone but in different times (1-3 days I guess, since it is impossible to keep synchronized 35 clonal lineages. I guess this is what authors mean to say. A diagram will improve that issue.

Other issue. When comparing in the discussion Cu EC50s across studies. Authors have to take into account that Cu suffers from a complex speciation in fresh water, This means that usually only a very small fraction of the total Cu use in experiments (i.e. 3%) is bioavailable to animals.

This means the very minor variation in the water used in different studies may affect dramatically EC50 for Cu. This is probably the reason justifying the apparently higher tolerance to Cu found in this study when compared with previous ones (Barata et al 1998).

It is also interesting to see that apparently deltamethrin induced greater changes in tolerance than chlorpyrifos. This may be related to the chemical properties and mode of action of both compounds and on the microevolutionary consequences. Pyrethroids have a KWO much greater than chlorpyrifos, which means that they can be adsorbed to particles and hence be more persistent in the environment (in sediments, edible food, etc). Chlorpyrifos just kill Daphnia, whereas pyrethroids at sublethal concentrations dramatically impair reproduction. This means that at low environmental concentrations pyrethroid insecticides are likely to affect to a greater extent Daphnia since they produce effects at lower concentrations than just chlorpyrifos and are more environmental persistent than chlorpyrifos. This is however, a conjecture, that need to be tested measuring pesticide levels on the sites and around them where Daphnia was collected

Author's Response to Decision Letter for (RSPB-2021-1903.R0)

See Appendix A.

RSPB-2021-1903.R1 (Revision)

Review form: Reviewer 2

Recommendation

Accept as is

Scientific importance: Is the manuscript an original and important contribution to its field?
Excellent

General interest: Is the paper of sufficient general interest?

Excellent

Quality of the paper: Is the overall quality of the paper suitable?

Excellent

Is the length of the paper justified?

Yes

Should the paper be seen by a specialist statistical reviewer?

No

Do you have any concerns about statistical analyses in this paper? If so, please specify them explicitly in your report.

No

It is a condition of publication that authors make their supporting data, code and materials available - either as supplementary material or hosted in an external repository. Please rate, if applicable, the supporting data on the following criteria.

Is it accessible?

Yes

Is it clear?

Yes

Is it adequate?

Yes

Do you have any ethical concerns with this paper?

No

Comments to the Author

The authors addressed all reviewer comments so improved the Ms that was already good

Decision letter (RSPB-2021-1903.R1)

19-Oct-2021

Dear Ms A. Almeida

I am pleased to inform you that your Review manuscript RSPB-2021-1903.R1 entitled "Differential local genetic adaptation to pesticide use in organic and conventional agriculture in an aquatic non-target species" has been accepted for publication in Proceedings B.

The referee(s) do not recommend any further changes. Therefore, please proof-read your manuscript carefully and upload your final files for publication. Because the schedule for publication is very tight, it is a condition of publication that you submit the revised version of your manuscript within 7 days. If you do not think you will be able to meet this date please let me know immediately.

To upload your manuscript, log into <http://mc.manuscriptcentral.com/prsb> and enter your Author Centre, where you will find your manuscript title listed under "Manuscripts with

Decisions." Under "Actions," click on "Create a Revision." Your manuscript number has been appended to denote a revision.

You will be unable to make your revisions on the originally submitted version of the manuscript. Instead, upload a new version through your Author Centre.

1) A text file of the manuscript (doc, txt, rtf or tex), including the references, tables (including captions) and figure captions. Please remove any tracked changes from the text before submission. PDF files are not an accepted format for the "Main Document".

2) A separate electronic file of each figure (tiff, EPS or print-quality PDF preferred). The format should be produced directly from original creation package, or original software format. Please note that PowerPoint files are not accepted.

3) Electronic supplementary material: this should be contained in a separate file from the main text and the file name should contain the author's name and journal name, e.g. authurname_procb_ESM_figures.pdf

All supplementary materials accompanying an accepted article will be treated as in their final form. They will be published alongside the paper on the journal website and posted on the online figshare repository. Files on figshare will be made available approximately one week before the accompanying article so that the supplementary material can be attributed a unique DOI. Please see: <https://royalsociety.org/journals/authors/author-guidelines/>

4) Data-Sharing and data citation

It is a condition of publication that data supporting your paper are made available. Data should be made available either in the electronic supplementary material or through an appropriate repository. Details of how to access data should be included in your paper. Please see <https://royalsociety.org/journals/ethics-policies/data-sharing-mining/> for more details.

<http://datadryad.org/submit?journalID=RSPB&manu=RSPB-2021-1903.R1> which will take you to your unique entry in the Dryad repository.

Once again, thank you for submitting your manuscript to Proceedings B and I look forward to receiving your final version. If you have any questions at all, please do not hesitate to get in touch.

Sincerely,

Dr Locke Rowe

Reviewer(s)' Comments to Author:

Please add your Dryad DOI to the Data Accessibility section.

Referee: 2

Comments to the Author(s)

The authors addressed all reviewer comments so improved the Ms that was already good

Decision letter (RSPB-2021-1903.R2)

22-Oct-2021

Dear Ms A. Almeida

I am pleased to inform you that your manuscript entitled "Differential local genetic adaptation to pesticide use in organic and conventional agriculture in an aquatic non-target species" has been accepted for publication in Proceedings B.

Data Accessibility section

Open Access

Paper charges

Sincerely,
Editor, Proceedings B
<mailto:proceedingsb@royalsociety.org>

Appendix A

To the editor of Proceedings of the Royal Society B

In response to the comments of editor and reviewers on the manuscript ID RSPB-2021-1903

Dear Dr Locke Rowe,

We thank you and the reviewers for the constructive feedback and valuable comments on our manuscript. Herewith we send our replies to all the comments from the reviewers and a revised version of the manuscript. The comments (presented in bold) and corresponding replies are listed in the order of appearance in the reviewers comments. Excerpts copied from the main text are presented in italic.

Replies in response to comments of Referee 1

Figure 1: I infer that the Y-axis values are predicted probabilities of immobilization. This should be specified.

We confirm the response variables are predicted probabilities. The figure has been adjusted according to the reviewer's suggestion (see below). We adjusted the caption accordingly.

Figure R1 (Figure 1 in the main text) – Logarithmic concentration ($\mu\text{g/L}$) – response (probability of immobilization at 48h) curves (A,C,E) and Average EC_{50} values (backtransformed) \pm C.I. (B,D,F), as assessed in a common garden rearing experiment using *D. magna*, for deltamethrin (DTM; panel A,B), copper sulfate (CS; panel C,D), and chlorpyrifos (CPF; panel E,F) for each land use type (nature reserve: diamonds, conventional agriculture: triangles, organic agriculture: circles) in which the ponds, from which the *D. magna* study populations were isolated, were located. Shaded areas in the left panels corresponds to 95% C.I. Panels represented in color correspond to the pesticides allowed in organic agriculture (DTM and CS), and panels in grey correspond to pesticide allowed in conventional agriculture (CPF).

Line 341: Not clear that mitigation is the appropriate term here.. Mitigating persistence of a population would seem to mean reducing the likelihood the population persists.

We agree with the referee and changed the text (Line 325 of the revised version of the manuscript): *“While rapid evolution in D. magna thus can enable their persistence in landscapes...”*

Replies in response to comments of Referee 2

I recommend to include in the SI a figure explaining the experimental set up. The typical ones showing populations, clones within populations, replicated within clones and how from each replicate test were performed. It is not clear to me the paragraph that describe that. If I am not wrong the authors mention that they perform acute test from each of the three replicates from each clone but in different times (1-3 days I guess, since it is impossible to keep synchronized 35 clonal lineages. I guess this is what authors mean to say. I diagram will improve that issue.

We followed the suggestion of the reviewer and included figure R2 in SI (now Figure S1 in the Supplementary Information). The tests were indeed spread through time, as it would not be feasible to do them all simultaneously. This was done by using parthenogenetically produced offspring of the 2nd to 5th clutch of each clone x replicate x pesticide rearing unit. During the rearing process, animals from different clutches were isolated in different jars to create separate replicates for each clonal lineage. Different concentrations of the pesticides were tested in random order for each clone and replicate, spread in time based on the availability of juveniles per clone.

Reference to the figure has been included in the main text (Line 136), and the reference to the other figure in SI (now figure S2) has also been adjusted (Line 307).

Figure R2 (Figure S1 in Supplementary information) – Experimental setup. Sensitivity of five clonal lineages for each population (two populations in nature reserves, 3 in conventional agriculture and 2 in organic, $n = 7$ populations * 5 clones = 35) was tested for chlorpyrifos, deltamethrin and copper sulfate. Seven

concentrations and a solvent control were tested for each pesticide. There were three replicates with each five individuals for each clone x pesticide x concentration combination.

When comparing in the discussion Cu EC50s across studies. Authors have to take into account that Cu suffers from a complex speciation in fresh water, This means that usually only a very small fraction of the total Cu use in experiments (i.e. 3%) is bioavailable to animals. This means the very minor variation in the water used in different studies may affect dramatically EC50 for Cu. This is probably the reason justifying the apparently higher tolerance to Cu found in this study when compared with previous ones (Barata et al 1998).

We thank the reviewer for this comment. We agree that variation in bioavailability could offer an explanation for why we find responses at higher concentrations when compared to other studies. Therefore have now added a sentence in the discussion to also accommodate this possibility (Line 278 - 281) *“One important aspect to consider is that copper shows speciation in freshwater environments (De Schamphelaere et al., 2003, Janssen et al. 2003, Winner et al. 1985). This can affect the bioavailability of copper in the water column. Small variations in water chemistry between studies, such as pH, can affect this speciation process, and this can lead to differences in sensitivity (Barata et al., 1998).”*. While we now added this alternative explanation, we think that the historical exposure of the populations to copper in Belgium entails a strong potential for adaptation, and therefore we still include this argument in our text.

It is also interesting to see that apparently deltamethrin induced greater changes in tolerance than chlorpyrifos. This may be related to the chemical properties and mode of action of both compounds and on the microevolutionary consequences. Pyrethroids have a KWO much greater than chlorpyrifos, which means that they can be adsorbed to particles and hence be more persistent in the environment (in sediments, edible food, etc). Chlorpyrifos just kill Daphnia, whereas pyrethroids at sublethal concentrations dramatically impair reproduction. This means that at low environmental concentrations pyrethroid insecticides are likely to affect to a greater extent Daphnia since they produce effects at lower concentrations than just chlorpyrifos and are more environmental persistent than chlorpyrifos. This is however, a conjecture, that need to be tested measuring pesticide levels on the sites and around them where Daphnia was collected

Our results do seem to indicate that deltamethrin can produce responses in *D. magna* at much lower concentrations compared to chlorpyrifos. We need, however, to be careful when engaging in a direct comparison of the magnitude of the responses to the two pesticides, as we tested responses to the pesticides in separate time blocks (as it would not be feasible to test 7 concentrations plus one control of three pesticides simultaneously). We opted for this strategy as our main goal was to compare the response of the different populations to each pesticide, not to compare the toxicity of the different pesticides. Hence an experimental design in which each pesticide was used at the same time was not required. While we do not anticipate any major differences between blocks of pesticides tested, we want to avoid making direct comparisons between pesticides. For this reason we prefer not to develop this comparative analysis in our text.

Other adjustments

The reference style has been converted to Vancouver, as this is the style used by the journal.

Data accessibility

A section on data accessibility has been added (Line 338 – 339 of the revised version of the manuscript). A corresponding reference has been added.

Data are made available at Dryad Digital Repository.

Acknowledgements

A section on acknowledgments has been added (Line 350 – 354 of the revised version of the manuscript).

We thank the editor and two anonymous reviewers for their constructive comments and feedback on an earlier version of the manuscript. We thank Edwin van den Berg for culturing the algae and Reindert Ekelson and Filipe Pinto for the valuable help with practical work. We thank our partners in the Belspo project ORCA for logistic support.